# Violence and Post-National Costa Rican Identity in *Limón Reggae*

**Anne Marie Stachura**

Department of Spanish and Linguistics, Franklin and Marshall College, Lancaster, PA 17603, USA;
astachur@fandm.edu

**Abstract:** Anacristina Rossi's novels have received critical attention relating to their presentation of pan-Caribbean identity and other challenges to the mythical national identity of the tico. Building on scholarship by Manzari and Kearns, I argue that *Limón Reggae* presents a representation of the post-national community and that the violent conditions that mark the protagonist's life not only debunk the national myth of a peaceful Costa Rica, but also comment on the impossibility of belonging in the post-national community. The pain that the protagonist experiences as a result of her interpersonal relationships reflects the difficulty of forming a community after the bounds of the nation have become less defined by globalization, even to individuals who come from groups not traditionally included in the definition of a Costa Rican citizen, such as the protagonist. With the breakdown of categories of affiliation across lines of geography, race, language, and class, the protagonist is able to move easily between places and groups, but her encounters with 'others' are complicated by the post-national condition.

**Keywords:** Costa Rica; global community; violence; 21st-century novel; Latin American literature

Anacristina Rossi's novels have received critical attention relating to their presentation of pan-Caribbean identity and other challenges to the mythical national identity of the tico. Her early novel, *La loca de Gandoca* (1996), "is an exposé of problems in Costa Rica: alcoholism, sexism, and the corruption of elected officials. It is a demythification of Costa Rica as a paradise, "green" or otherwise" (Barbas-Rhoden 2011, p. 128). Manzari considers Rossi's *Limón Blues* (2002) and *Limón Reggae* (2007) to be part of a literary movement rewriting the history and present condition of Limón, stating, "A través de una verdadera estética de resistencia presente en la escritura contemporánea de la costa atlántica costarricense, los escritores de la región establecen un auténtico contrapunto entre lo que se podría llamar la comunidad imaginada de la nación y la real de la región limonense" (Manzari 2007, p. 254). At the same time, *Limón Reggae* presents Limón as firmly interconnected to the global community. Starting from Sommer's *Foundational Fictions*, Kearns observes that the romantic attachments in Rossi's work are metaphorically linked not to the national but to the post-national condition, writing, "As nineteenth-century Latin American *romances* had done, erotic liaisons in *Limón Blues* had represented allegorically new national alliances that crossed class and racial lines, in this case between West Indian and Central Valley Costa Ricans. *Limón Reggae's* liaisons go a step further erasing national boundaries" (Kearns 2007). Building on Manzari and Kearns, I argue that *Limón Reggae* presents a representation of the post-national community and that the violent conditions that mark the protagonist's life not only debunk the national myth of a peaceful Costa Rica, but also comment on the impossibility of belonging in the post-national community. The pain that the protagonist experiences as a result of her interpersonal relationships reflects the difficulty of forming a community after the bounds of the nation have become less defined by globalization, even to individuals who come from groups not traditionally included in the definition of a Costa Rican citizen, such as the protagonist. With the breakdown of categories of affiliation across lines of geography, race, language, and class, the protagonist is able to

move easily between places and groups, but her encounters with 'others' are complicated by the post-national condition.

The unique cultural identity of the *limonense* community within Costa Rica and its relationship to the nation have been marked by oppression, a condition that is continued in the representation of members of this community in the post-national imaginary of *Limón Reggae*. Limón's Afro-Costa Rican, English-speaking population challenges the notion of a Costa Rican national identity that is Spanish-speaking and ethnically of Spanish descent. Gringberg Pla notes that the *limonense* community has a complicated relationship to Costa Rican national identity and has found a transnational Afro-Atlantic identity that is reflected in Calypso music (Gringberg Pla 2012, pp. 405–6). Similarly, in her study of music and identity in the novel, Sophie Large (2020) observes that music serves as a "seña de identidad para la comunidad afrolimonense, contectando con otros componentes identitarios fundamentales, tales como la lengua y la religión. Segundo, y al mismo tiempo, sirve de refugio para los miembros de la comunidad afrolimonense frente a una realidad marcada por la opresión" (4). While seemingly the erosion of the concept of a national imaginary would make the lack of belonging irrelevant, this is not the case in the post-national imaginary. Nor do the oppression and resistance to oppression end with the weakening of the nation-state. The term post-national imaginary does not mean that the nation-state and nationalism are no longer relevant; indeed, as Benhabib and Resnik (2009) note, "even porous borders continue to define networks of obligations and constitute real barriers, rendering some persons aliens or intruders" (Benhabib 4). Rather, the post-national imaginary presents an image of the global community, wherein "subject to fewer restrictions and greater speedup, the circulation of people, capital, and messages brings us into daily contact with many cultures; consequently, our identity can no longer be defined by an exclusive belonging to a national community" (García Canclini 2001, p. 91). It is precisely this rapid exchange that creates conflict within and between nation-states. In his essay "The Fear of Small Numbers", Arjun Appadurai explains that "the speed and intensity with which both material and ideological elements now circulate across national boundaries have created a new order of uncertainty in social life" (Appadurai 2006, p. 5) and warns that "where one or more of these forms of social uncertainty come into play, violence can create a macabre form of certainty and become a brutal technique (or folk-discovery procedure) about 'them' and, therefore, about 'us'" (Appadurai 2006, p. 6). In this global community, the increased interaction with the other leads to a propensity to commit acts of intimate and interpersonal violence, which is the situation of the protagonist of *Limón Reggae*.

In the post-national Costa Rica of the novel, identifying allies and a potential community is fraught. This emphasis on identity and identifying others is evident in the opening sentences of *Limón Reggae*. The novel opens with a description of the city of Puerto Limón, Costa Rica, through the perception of the protagonist, Laura, who in 1971 is sixteen years old. The narrative voice notes that Laura is attentive to the physical attributes of those who surround her: "le fascinan los detalles, una sonrisa con los dientes de adelante muy separados, un tono de piel tan negro que parece azul" (Rossi 2008, p. 9). In particular, Laura is fascinated by faces and eyes, and visual expressions and vision become an overarching theme in the novel. In the exposition, Laura's racial identity and its manifestation in her facial features are explored. Laura's mother is Lebanese, and her father is of mixed race (black and white). In an analepsis, the narrator describes Laura's self-perception while she is living in the capital city, San José, with her parents, through an examination of her face: "Mira su boca no muy grande, como dibujada, le gusta su boca, sus labios parecen a los de su papá", and mentions that "Le importan sus ojos . . . Piensa que fue por sus párpados que una vez una compañera nueva la señaló y dijo burlona "¡Esa es turca!" (Rossi 2008, p. 11). Laura's face sets her apart from her peers, particularly her eyes. Both her racial identity and her ethnic heritage are evident physically and are perceived by those around her. She is powerless to hide these aspects of her person; however, her unique composition both hinders and benefits her in different situations as she moves across national and political

lines, throughout Costa Rica and to El Salvador, Nicaragua, and the United States during the 30-year time frame spanning from the end of the 20th century to the beginning of the 21st.

Just as Laura is ambivalent about identifying with a particular race or locality even within her homeland, throughout the novel she consistently changes her name as well. As a child, the individuals who are closest to her, her aunt and her best friend, call her Aisha: "Sólo dos personas en este mundo le dicen Aisha o Habibi: Percival y Maroz" (Rossi 2008, p. 28). She later uses Aisha as her code name while fighting in the revolution; she is known as "Compañera Aisha". After losing her family (specifically her adopted son, Toño) in the violence of the revolution, she changes her name once more to Anaís, yet she is never fully comfortable being identified with this name: "Lleva ya tiempo llamándose Anaís pero aún se sobresalta, se queda un segundo perdida" (Rossi 2008, p. 130). Later, she tells a lover she meets in El Salvador that she is known by yet another name and identity, saying, "En realidad yo no soy Aisha. Soy la compañera Salomé. He viajado por todo el mundo, por eso hablo otros idiomas. He estado en Francia, en Italia, en Estados Unidos" (Rossi 2008, p. 162). The use of various names (nicknames and code names versus her "official" birth certificate name) reflects the instability of the identity of the protagonist in *Limón Reggae*. Laura/Aisha/Habibi/Anaís/Salomé is presented throughout as a complicated, multifaceted individual; she cannot be identified by only one name, nor does she define herself by belonging to a particular racial group, national body, familial structure, or social class division. She moves fluidly (like the air and water metaphors that describe her emotional states) across racial, national, and class boundaries throughout the narrative.

Just as Laura's face and eyes reveal much about her background, her observations of the faces of those around her reveal much about her perception of others' motivations and feelings. At a pivotal moment in her youth, Laura is uprooted from her safe community and neighborhood due to her family's financial circumstances and is exposed to extreme poverty and degradation, which leave her traumatized. Once again, her change in circumstances—the fluidity with which she moves from one social situation (a stable home life in an economically advantaged area of the city to a deprived area of the city) results in a painful experience that produces a feeling of malaise in her character. The rapidity of this change and the feeling of being overwhelmed by this sudden adjustment are reflected in the rhythm of the text, which uses accumulation and polysyndeton to reflect the protagonist's reaction to her new circumstances: "Esa era su vida, una vida normal hasta que su papá dejó el Banco de Seguros y se metió a hacer negocios y lo engañaron y lo dejaron debiendo unos platales que nunca iba a poder pagar y la vida entera de Laura se derrumbó. Lo perdieron todo y tuvieron que alquilar una casa espantosa y chirrisca en un barrio fatal y su papá se metió en una cama y ya no quiso salir y su mamá se puso a coser ajeno y a cocinar para vender y no quedó de la normalidad de Laura piedra sobre piedra." (Rossi 2008, p. 14).

In her new living situation, Laura witnesses acts of cruelty and humiliation that fascinate and repulse her. She is traumatized by the behavior of the impoverished unsupervised children who live in the slums of San José, and her lasting impression is of their facial expressions. Laura watches the children defecate in public as the narrator describes Laura's reaction, stating, "Pero lo que se le convierte en una obsesión es la cara que ponen. La boca se les deforma, los labios se les van para abajo, retadores y crueles. Es una cara de placer y al mismo tiempo despiadada, feroz, una cara adulta con lo peor de los adultos, que Laura presiente pero no sabe qué es" (Rossi 2008, p. 14). Due to her inability to put a name to this phenomenon, Laura denominates this expression as having "eso" or "algo", which she returns to at various moments throughout the novel. The failure of language to describe this cruelty is significant and is referenced later in the narrative.

During Laura's formative years, she is exposed to "eso" or "algo" in relation to acts of cruelty and humiliation that do not involve subjective violence toward other human beings. In addition to public defecation, the neighborhood children also torture small animals, such as cats: "Los agarran y les arrancan una pata o una mano, los descuartizan,

o les estallan los ojos con palos y clavos, Laura no puede dormir porque pasan las horas y sigue oyendo los agónicos maullidos" (Rossi 2008, p. 15). Laura compares this behavior to a science experiment in which she is made to dissect a frog; in her view, to kill in the name of science is evidence of a lack of respect for life. This incident is an early indication of her doubt of science in particular and of authority in general. In her attempt to make sense of this cruelty, she tries to extend the metaphor to determine which 'Professor' would have ordered the neighborhood children to experiment on innocent animals. She had only seen "eso" or "algo" in the faces of children. It is only later that she first witnesses the expression on the face of an adult man, when an upper-class woman driving a luxury car hits a pedestrian crossing the street in the slum, and although there is no significant damage, the man becomes enraged and begins hitting the car with a brick. This scene prefigures further incidents involving class conflict and "eso" or "algo" in *Limón Reggae*.

Class, politics, and race are the primary considerations in the narrative as Laura becomes a revolutionary and begins working to organize the people of Limón. Even in this capacity, she continues to observe faces, eyes, and expressions to judge character and motivation. While working to organize a massive strike in the city of Puerto Limón on the Caribbean coast of Costa Rica, she accompanies a fellow organizer, Jorge Cole Brown, and is alarmed by the look of the crowd assembled there; the narrator first notes their extremely thin builds and then observes, "Y las caras, piensa Laura. No tienen "eso" pero tampoco tienen expresión, son máscaras impasibles, sus ojos no brillan. Jorge Brown está loco si cree que es posible motivar a esta gente" (Rossi 2008, p. 109). However, she misjudges the motivation of the masses; when the strike takes place, Laura sees hatred in the eyes of the people and comes to an understanding about revolutionary violence: "los ojos de los suampos ya no son canicas inertes, ahora tienen una chispa, algunos tienen fuego, es como un fuego lento, piensa Laura. Y entonces intuye que estas luchas sólo se ganan con odio y ese fuego es el odio y el fin de la paciencia o de la pasividad" (Rossi 2008, p. 114). The hatred that Laura sees in the eyes of the protesters inspires violence against the system, which she considers to be positive; while earlier she had described their eyes as being lifeless and therefore their faces as mask-like, during the strike she sees them as a powerful force. At the same time, those who are charged with controlling the crowd and breaking up the strike are also associated with masks; on the third day of the strike, Laura's aunt Maroz calls her to the window and indicates, "Laura vení, mirá esas máscaras y esos grandes escudos como los de los cruzados" (Rossi 2008, p. 116). In the service of the Repressive State Apparatus, the masks serve to provide anonymity and protection for the police force, erasing individuality and making them all "uniform". Due to the mask, it is impossible to determine what motivates each individual, either "eso"/"algo" or the class hatred of the protesters. Throughout the novel, Laura relies on her ability to determine individual motivations through eyes and facial expressions in order to distinguish friends from enemies, yet she continues to encounter "masks" that preclude her from making a positive identification.

By the end of the novel, Laura has come to a conclusion about humanity, cruelty, and violence. Upon her return to her native Costa Rica, Laura is confronted almost immediately with violence. While witnessing the brutal murder of an old woman perpetrated by gang members in San José, she makes eye contact with the young man who had committed the crime: "Y en su expresión ella vio "eso", vio "algo", vio la degradación que la obsesionaba desde la infancia" (Rossi 2008, p. 283). After this incident, Laura is kidnapped and forced to live with the gang members for an extended period of time. During this time, she determines that her initial conclusion about degradation and violence was inaccurate; she tells a friend, "Siempre había pensado que "eso" era una deshumanización. Pero no. Ahora sé que "eso" es lo más humano que hay, no existe entre los animales" (Rossi 2008, p. 299). The capacity for cruelty that had always fascinated the protagonist is something that she uses to define humanity by the end of the novel, given her experiences in the post-national imaginary. Laura is horrified by this phenomenon, since she is constantly trying to understand others by reading their eyes and facial expressions. Through her

experiences in the slums of San José and with the gang members, she comes to witness the violence that proves just how futile her attempt to know the other is; as Elaine Scarry theorizes, "the ease of inflicting injury (as well as the omnipresence of the impulse to injure) shows the difficulty of knowing other persons. There exists a *circular relation* between the infliction of pain and the problem of otherness. *The difficulty of imagining others is both the cause of, and the problem displayed by, the act of injuring*" (Scarry 1996, p. 102). The subjective violence presented in the novel is evidence of the difficulty of imagining the other, a primary concern in the global community presented in *Limón Reggae*.

If indeed Laura's eyes mark her as an outsider socially, her vision allows her to gain admittance to both elite and grassroots circles. While an art student at university, Laura is exposed to the lifestyle of upper-class Costa Ricans as a result of her artistic talent; she is invited to a party where she meets Luís, who asks her to go out with him, and since "Laura quería conocer la alta burguesía bohemia", she accepts. (Rossi 2008, p. 133). Through Luís, Laura is introduced to high bourgeois bohemian habits, such as smoking marijuana and drinking excessively at parties, but what she most appreciates is visiting their beach houses: "Sin Luís Laura jamás hubiera conocido esas extrañas residencias que estimularon su arte. Los sorprendentes detalles de las piscinas, de los salones, de los balcones cortados a pico sobre los acantilados atraparon tanto a Laura que se puso a pintarlos en la forma más fiel y descubrió así su talento para el hiperrealismo. Se puso a imaginar puntillosa lugares internos, lugares imposibles que al pintarlos se volvían totalmente reales. Ya descubrí mi estilo, pensó." (Rossi 2008, p. 133).

However, crossing the boundary into the lifestyle of the upper class is a transition that is also painful for Laura. Ultimately, Laura rejects both Luís and his social circle after witnessing his friends' abuse of the hired help in a drunken and drugged stupor, but the hyperreal painting style she discovers stays with her throughout the entire narrative and allows her economic independence and even a small amount of fame. After rejecting Luís, Laura commits herself fully to radical politics and to fighting in the revolution, where she sees new traumatic scenes of suffering and violence; upon her return to Costa Rica, she channels her memory into art: "Compra lo necesario y en el cuarto pequeño de la casa de sus padres se pone a pintar. Pinta unos cuadros terribles que no le van a comprar las señoras burguesas, cuadros hipperrealistas de cuerpos mutilados en paisajes sublimes, soleados, vibrantes" (Rossi 2008, p. 200). The explicit contrast between the peaceful, sunny landscapes and the violent mutilated corpses is meant to repel the bourgeois patron; indeed, as Sontag explains, "there is shame as well as shock in looking at the close-up of a real horror" (Sontag 2003, p. 42). The narrator implies that Laura is conscious of her decision to purposefully illustrate her experience of the war without regard for the possible commercial uses of her art. Later, she uses her hyperrealist style to capture the beauty of Puerto Viejo, Limón, in a series of paintings that receives critical praise and is a financial success for her. In the series, she explores the effects of light and shadow, her inspiration coming from the fact that "Nada dura en el sur de Limón" (Rossi 2008, p. 270). The temporality of life in Limón that Laura expresses in this statement is reflective of the region's history of immigration; according to Mosby, "West Indian workers never intended to stay in Central America, and a culture of temporality was constructed. Life in Limón for many West Indian workers and their families was founded on the impermanence of the situation" (Mosby 2002, p. 12). Additionally, Laura's life choices reflect this impermanent situation only in reverse, for she immigrates first from Costa Rica to El Salvador, and later to New York, only to return to Puerto Limón at the end of the novel.

Upon her return, a friend asks Laura about Puerto Viejo and her artwork in a dialogue that comments on attachment and commerce: "Aisha abre los párpados, sonríe con tristeza, 'Ahora el Puerto Viejo lindo sólo existe en mis pinturas', dice. '¿Y dónde están esos cuadros?' pregunta Mai. 'No sé, en casa de particulares, los vendí'" (Rossi 2008, p. 296). Despite her attempt to fight for social justice during the revolution in El Salvador, Laura is aware of the persistence of inequality; although she has created cultural artifacts that capture the majesty and history of Puerto Viejo, these items become private property only to be enjoyed by

those who can afford to possess them. The protagonist's sadness stems not from a loss of the physical objects as much as the acceptance of the persistence of injustice. The unchanging part of Laura's identity throughout the novel is her commitment to social change and social justice; whether working with Black Radical groups in Limón, revolutionaries in El Salvador, or even as a translator for the U.N. in New York, she remains preoccupied with the need to redistribute wealth both in her native Costa Rica and in the other countries she inhabits. At the end of the novel, Laura's only attachment is to the ideal of social justice.

Anacristina Rossi's presentation of a mixed-race, multilingual protagonist who engages in revolutionary violence problematizes the construction of Costa Rican national identity. As Dorothy Mosby observes, "The Central American republic of Costa Rica is famous for its national myths: its natural tropical beauty, the mystical qualities of its coffee, the abolition of the armed forces, and its democratic stability in a region of political violence . . . myth in this instance is not a compendium of tales of national origin, but rather a cultural fiction diffused and repeated to convey a set of cultural values. This set of beliefs is held by members of the nation and defines their identity as a people." (Mosby 2002, p. 23).

The myth of the peaceful and egalitarian Costa Rica is challenged by Laura's participation in radical movements. While studying art at the university in San José, Laura confides to a close friend that she has stopped going to class, due to the imminence of the revolution, to which the friend replies, "Oíme, Laura, estás loca, ¿cómo vas a hacer una revolución en un país donde el mismo gobierno propone la reforma agraria? La revolución es para los otros países centroamericanos, donde los ricachones no ceden ni pizca. Nuestro país es diferente" (Rossi 2008, p. 98). However, this myth of Costa Rican exceptionalism is debunked when in a dialogue between the President and his advisor; after realizing that public pressure will not allow him to introduce any lasting social change, he states his new plan: "voy a crear entidades estatales para impulsar el desarrollo. Eso no molesta a los ricachones, al contrario. Voy a universalizar el seguro social. Eso calma a los agitadores y no saca roncha. –¿Con qué plata, Dionisio? –pregunta Mai. –Esa es la diferencia entre don Pepe y yo, Mai. Yo tengo la plata de los banqueros internacionales que no saben qué hacer con sus petrodólares" (Rossi 2008, p. 107). The implication that the tranquility of Costa Rica is due not to progressive government policies and reform but rather to an influx of international aid dollars undermines the image of a socially cohesive nation that stands as a counterpoint to other Central American countries. Despite this disenchantment, Laura continues to participate in radical politics.

Laura's inclination towards participation in political movements for social change begins as a child. In an analepsis, Laura remembers watching the film *Lawrence of Arabia* in Limón with her Arab-Costa Rican aunt, Maroz, and their Afro-Costa Rican friends, Samuel and Percy. As they play after watching the film, the young Laura declares to her childhood friend Percy, "¡Yo soy Awrence y vengo a sacar a los turcos de Arabia! ¡Soy un guerrero, soy la guerrera Aisha!" (Rossi 2008, p. 35). Her first instruction in radical politics takes place later in Limón, where she attends classes through CoRev, a group co-founded by Percy. In deciding whether to stay in Costa Rica or to join the revolutionaries in El Salvador, Laura returns to Limón and to her aunt, who tells her that during her youth, she too had been a rebel and "que ella se transformó en su famoso viaje árabe y que por años colaboró con los rebeldes palestinos" (Rossi 2008, p. 136). This establishes a radical political lineage for Laura (which parallels Percy's grandfather's involvement with Garvey's movement, the U.N.I.A). Maroz tells her niece about her traumatic experience fleeing Jaffa for Jordan on foot, prefiguring Laura's own gruesome trials as a guerillera. Laura decides to go to El Salvador, and there her political genealogy allows her refuge; she goes to stay with a friend of her aunt who informs her that she has arrived at a crucial time, explaining "que ese país es el más injusto de América y el más extraordinario, que maestros, obreros, empleados públicos, profesionales, sindicatos, de todo, van a echarse a la calle muy pronto, en dos días" (Rossi 2008, p. 137). Laura's political commitment links her to all of those

who participate and have participated in land reform movements, including her family members (Maroz, in Palestine) and her co-revolutionaries in Limón and El Salvador.

Laura's political genealogy is her most salient defining characteristic, and in *Limón Reggae*, she is presented as part of a global community of radicals. However, this community of belonging is her only option, since Laura will always be marked as other; she is prevented from forming part of the tico community, for "Within the Costa Rican context, then, a national identity has been created so that those not adhering to the implicit rules of the tico are forced to either join the ranks of 'Otherness' or seek life in exile, both instances pointing towards living in a permanent state of non-belonging either within or outside the physical borders of the country" (Harvey-Kattou 2011, p. 22). In addition to her connection to her Aunt Maroz in Palestine, she is linked to her revolutionary contemporaries in El Salvador and those who fought for land reform in the past; a friend describes to her the situation in her newly (temporarily) adopted country, mentioning that "la rebelión de 1932 se dio porque le habían quitado a los indígenas y campesinos sus tierras ejidales y los habían convertido en seres misérrimos que vendían su mano de obra por una tortilla en las grandes fincas de café" (Rossi 2008, p. 141). Her connection to Percy also implicates his grandfather, a member of the Marcus Garvey movement, whose connections to land rights in Limón is detailed in Rossi's 2002 novel, *Limón Blues*, in which the protagonist loses the rights to his land to the United Fruit Company and struggles for economic survival thereafter. Her politics link her to individuals and groups across national and temporal boundaries (in Costa Rica, El Salvador, and Nicaragua, both in the past and at present) that are united in their struggles for land reform (principally) and social justice (generally). In the course of moving and struggling to achieve these goals, Laura leaves her family (Aunt Maroz, Percy and his grandfather in Limón, and her parents in San José) and attempts to establish meaningful relationships with others committed to a similar ideology; however, like the radical engagements themselves, Laura's intimate relationships are not as successful as she imagined, and involve a surprising amount of violence that comments allegorically on the state of the global community representing post-national Costa Rica.

While Laura/Aisha is in El Salvador with the other guerrilleros, she has the opportunity to begin a romantic partnership with one of her fellow revolutionaries, an alliance that she eventually rejects and that ends in extreme violence. Although, at this point, Laura has formed a lasting bond with Toño, her adopted son, she is hesitant to enter into a hegemonically normative, stable nuclear family. Her colleague Domingo makes a number of sentimental advances, to which Laura at first responds with a cool distance and later with an understanding of friendship. When the subject of a family is broached in a dialogue between the protagonist and her adopted son, Laura rejects the possibility as a fantasy: "Toño le dijo, susurrando: 'Mama, dice Domingo que se va a acompañar con vos y entonces va a ser como mi papá. ¿Eso es cierto?' 'No, eso no es cierto,' le respondió Aisha muy bajito. 'Pero él dice que es verdad', susurró Toño. 'Eso se llama una fantasía. Tiene derecho a tenerla'" (Rossi 2008, p. 163). Before any intimacy can be formed between Aisha and Domingo, there is an incident in which the revolutionaries are ambushed and it is revealed that Domingo is a traitor: "No sabían cuánto había revelado al ejército pero Domingo manejaba información de seguridad, las rutas de entrada, las rutas de salida, ubicaciones estratégicas" (Rossi 2008, p. 166). The inability to know the other and to distinguish between friends and enemies is evident in the relationship between Domingo and the other revolutionaries. Previously it was established that Domingo was different from the other revolutionaries due to his appearance and origin; the narrator describes that "[Domingo] Era de la ciudad y hablaba muy diferente a los campesinos. Algunos lo rechazaban por eso o talvez porque le ponía mucha atención al vestir" (Rossi 2008, p. 161). Laura's personal relationship with Domingo further comments upon this inability to connect with another human being, and the resulting violence is due to her failure to identify the other.

After the ambush, in which several of her comrades are killed, Laura exploits her sexual power and romantic attachment to Domingo to first seduce and subsequently murder him. For this project, Laura invents another identity; in a telephone conversation,

she tells her soon-to-be victim, "ya no soy Salomé, nunca más. Ni Aisha. Soy Roxana" (Rossi 2008, p. 168). To accompany her new name, Laura alters her appearance. Without revealing her murder plot, she asks her friend Reina to give her a makeover: "que por favor le ayudara a ponerse despampanante, que quería verse como femme fatale" (Rossi 2008, p. 168). Her transformation is effective and affords her some degree of anonymity from the outset, as her target cannot identify her due to her new look: "Al verla bajar del taxi Domingo da un respingo, no la reconoce, cree que es la mujer de algún alto oficial, pero las mujeres de los altos oficiales no llegan en taxi" (Rossi 2008, p. 169). During their reunion dinner, Domingo admits to his treachery and tries to justify his actions. After ensuring that Domingo has ingested enough alcohol to impair his judgment and drugging him by putting crushed pills into his digestif, she accompanies him to his house and suffocates him after he has passed out. In this case, the female body is presented as an "arma de resistencia y autonomía" (Pearson 2011, p. 126); however, after committing this violent act, Laura does not feel triumphant about having killed the traitor, realizing that he is just a part of a larger system of injustice within the country itself: "Ella sabía que sería así, que no tendría remordimiento, pero no contaba con esta amargura. La amargura es saber que Domingo, traidor y canalla no es más que un pobre diablo. Un títere en un tinglado. Y el dueño del tinglado no es el ejército ni Estados Unidos sino alguien que Aisha vio ayer" (Rossi 2008, p. 173), referring to a young wealthy woman she encountered in the street who was being driven in an armored car. Laura feels sorry for Domingo and for all of the poor who are corrupted by the rich and become traitors to their comrades. She must kill Domingo because of his treason, and it is the social structure of the world that they inhabit that prohibits their successful "fantasy" union.

In addition to her failed romantic relationships with Domingo and Luís, Laura also has an unsuccessful parent–child relationship with her adopted son Toño, which ultimately becomes violent and manipulative as well. Laura unofficially adopts the child after he is orphaned when his family is killed by soldiers during a military operation. Even before Toño's parents and brother are confirmed dead, Laura becomes the child's caretaker and surrogate parent; the narrative voice mentions, "Seguían sin saber nada de Antonio y de Sonia. ¿Estaban vivos, muertos, dónde estaban? Toño había empezado a llamarla mama" (147). Laura and Toño survive intense combat situations, a lack of food and water, and serious illness and injury together in El Salvador, thereby making their connection stronger and more intimate. Laura's acceptance of her maternal role comes gradually. Upon returning from a trip, "Toño se echa a los brazos de Aisha suspirando '¡Mama!' Aisha lo abraza, besa sus manos flacas, ásperas y sucias. Nunca en su vida pensó en tener hijos y ahora tiene a ese cipote" (185). Toño becomes a messenger for the guerrilla and one day does not return from a mission; Laura assumes he is dead because "encontraron un desecho que parecía ser su hijo diminuto. Los chuchos le habían destrozado la cabeza y cortado los pies" (197). Despite her inability to confirm the identity of the cadaver as her own child, Laura is once again decentered by this loss, as is evidenced by the description of her actions; the narrator says that after learning of Toño's death, she begins to *rodar*, or "Ir de un lado para otro sin fijarse o establecerse en sitio determinado" (Rodar 2022, DRAE). The violent end of her relationship with Toño results in Laura's further loss of stability and encourages her wandering and constantly mobile lifestyle.

Laura's penchant for relocation takes her to the United States to work as a translator. While living in New York after the September 11 terrorist attacks on the World Trade Center in 2001, Laura reencounters Toño, whom she had assumed to be dead. As a witness of gang violence, she is accosted by two gang members, one of whom she recognizes as her long-lost son. Instead of killing her, the gang kidnaps her and takes her to live with them. After escaping from the gang and returning to Costa Rica, she continues to defend Toño, despite the violent and destructive acts she witnesses; a friend, doubtful about the supposed reunion and suspicious of Stockholm syndrome, asks her how she was certain that the young gang member was Toño, to which Laura explains: "Porque una noche se puso a peinarme y se durmió con mi pelo en la mano como cuando niño. Pero se arrepintió,

y al día siguiente me rapó la cabeza. Y un día me pidió que me tatuara. Me tatuaron los pechos. No, nada más los tatuadores, él, y yo. Y desde ese día permitió que lo llamara Toño" (300). Although Toño does not go so far as to kill Laura, his mother, the actions of maiming her body with tattoos and shaving her head indicate the inability of the two characters to maintain a mother–son relationship that does not involve physical violence and intimidation. Therefore, Laura is unable to develop a male–female relationship that is not marked by violence or exploitation throughout Rossi's novel.

The novel closes with Laura's return to Costa Rica and her failed attempt to help Toño leave the gang and start a new life. She is with her friends Mai and Federico from her revolutionary days, yet she has no stable familial or romantic relationships. The ending of the novel is open, as Laura and her friends set out on a dangerous mission to find Toño and stop his participation in the flow of illegal drugs or weapons across borders, a plan that has little hope of success. The final image of the novel is of the three characters driving off to find Toño in an armored vehicle, recalling her earlier encounter in the novel with the young woman in a similar car who symbolizes the problem of inequality for Laura. At the end, Laura is referred to as Aisha, but she is a defeated warrior. Her attempt to change problems of social injustice by joining a revolutionary group are futile, leaving her traumatized emotionally, and her effort to form meaningful intimate relationships leads invariably to physical or psychological violence. *Limón Reggae* presents a militant protagonist as part of the global community, and her experience reflects the impossibility of achieving social justice or personal happiness in the modern world system.

**Funding:** This research received no external funding.

**Institutional Review Board Statement:** Not applicable.

**Informed Consent Statement:** Not applicable.

**Data Availability Statement:** Not applicable.

**Conflicts of Interest:** The author declares no conflict of interest.

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
