# Peer review of "Violence and Post-National Costa Rican Identity in Limón Reggae"

_humanities, doi:10.3390/h12040056_

Round 1

Reviewer 1 Report

Comments for paper  “Violence and Post-National Costa Rican Identity in Limón reggae” in the Open Access Journal  MDPI Humanities

This is a very well-written paper and a good addition to the body of research on Rossi’s novel Limon Reggae.  It adds a different interpretation, demonstrating with plenty of evidence that Rossi offers no hope to the turmoil of Central America in the last decades of the twentieth century. 

My main suggestion is to directly address the contrast of this new interpretation to previous ones that suggest that the rhythms of reggae music are the coalescing element in the otherwise disjointed political and identarian situation that the novel portrays.  Especially the analysis by Valeria Grinberg Plat should be taken into consideration (“Ritmos caribeños, transnacionalismo y narrativa en Centroaméricana” Versiones de la modernidad.  Literaturas, identidades y desplazamientos.  Cortez, B, et al. Editors. Guatemala:  F & G Editores, 2012.  P.  393-414.). Considering the title of the novel and the importance of music within it, and previous interpretations by other literary critics, will make this paper’s argument stronger.

Other suggestions:

·      Several long quotes should appear in separate indented paragraphs (for example the quotes in between pages 2 and 3, and pages 4 and 5)

·      Add subtitles to different sections of the paper to make the paper’s reading easier. 

·      The conclusion seems a little abrupt.  Should include a brief summary of main points discussed.

Author Response

Many thanks for your careful reading and thoughtful recommendations to improve my paper. In my new version, I have incorporated Gringberg Pla's argument as well as other interpretations (Large, Pearson) that I had neglected to engage with in the previous draft.

Reviewer 2 Report

First of all this is an excellent paper! With this in mind the suggestions I have are only minor and should not take very long to do or detract from the overall flow of the paper.

Page1, line 31. There is a need to define what a post-national condition is as even though the novel does speak about the character crossing borders , the very fact that borders are being crossed and the nations are still named as Costa Rica, El Salvador, and the United States signifies that there is a national space, however porous. So, a definition or clarification as to what is meant by a post-national condition is in order. A "post-national" community is also mentioned in page 1, line36. I think the writers mean a diaspora or a transnational community of people who, though not caught between borders or because they are caught, no longer identify with either nation (origin or adopted). In any case this needs to be defined by looking into what the writers who use the term mean it such as Kearns. 

2. Page 5, line 239. The citation of Mosby's text states "Costa Rica is famous for its national myths: its natural beauty" etc. The citation is fairly long and uses ellipsis to abbreviate but it leaves out a part of the national myth that deals with race and this part is important as part of the reason why the main character and the people of Puerto Limon by their very existence as Costa Ricans that do not correspond to this myth challenge the "tico's identity. There is a need to elaborate further on this point alongside or without the long quotation from Mosby. 

3. Page 8, lines 395-6. Before the conclusion, it is important to talk about the role of music in this narrative or why Reggae and its connection to Limon Blues (2002). There is a brief mention of Limon's connection to what Paul Gilroy called The Black Atlantic and I think if the space allows it parsing this out briefly before as part of the conclusion will make the paper stronger. 

4. Page 9, line 435. The title of the work by Cohen is missing. 

There is only an awkward sentence but only because the quotation is in Spanish on page 4, line 184. The quotation is introduced by a very long sentence beginning with "While an art student...she is invited to a party where she meets Luis, who asks her to out with him, and since "Laura queria conocer la alta burguesia bohemia y por eso acepto." It needs to be broken down into smaller sentences and find a better way to introduce the quotation that is not awkward. 

Author Response

Many thanks for your careful reading and recommendations to improve my paper! I have incorporated your changes:

  1. There is now a paragraph that defines the theoretical framework of post-national imaginary explicitly, directly after the introductory paragraph.
  2. and 3. I added a section on race and national identity, specifically referencing criticism from Gringberg Pla and Sophie Large about music and afro-Atlantic identity that I had neglected in the previous draft.
  3. Thank you for noting the bibliography typo and for the suggestion to re-word the quote on page 4. I believe it is less awkward now in combining the English and Spanish.

Reviewer 3 Report

I was pleased to review the application of the Costarican colleague on the representation of that country in its current literature and narratives. As a matter of fact this is a new and fresh period of narratives on Costa Rica, not the classical of violence and repression but also the current  national notion of a social paradise, free of conflicts.

I found interesting, focused, and a new reading of Costa Rica as a mythical,

social, and political privileged space of peace, democracy, and community. This narrative

of social equality, democratic politics, and universal education in its modern version

of exceptionalism has keep the country united and open to tourism and with the same system. The

emigration to other countries and specially to the U.S. tells another story. Migrations to Spain and the U.S. tells another story. The complexity of the case is promising and deserve support.

Are you discussing authors living in Costa Rica as well as authors

self-exiled in the U.S. or Spain? 

Author Response

Many thanks for your careful reading of my paper and your comments. I agree that the question of migration and exile is a promising avenue for future research. I appreciate the time you took to help me improve my article.

Reviewer 4 Report

I would suggest to incorporate a more detailed and thorough historic context for the revolutionary Black /Indigenous movements in the area connecting El Salvador, Costa Rica and the USA.  A summary of the plot is lacking  so  the reader has  a difficult time following the article's interpretation.

It would be also advisable to elaborate more on the revolutionary multifaceted identity of the main character, particularly, regarding all the physical abuses she endured and how the race/class connections are played out. 

The analysis seems to be anecdotic in some passages of the article. It would be advisable to discuss the thesis about  a pan American identity in a post national context, offering clear definitions of the conceptual apparatus  used in the article. 

I would suggest to review the bibliography. There are some missing articles published about the novel and the author not considered, such as Carol Pearson or S. Large. 

Author Response

Many thanks for your careful reading of my article and for your suggestions to improve it. I have attempted to give a more specific timeline of the novel's plot to make the argument easier to follow. Thank you especially for calling my attention to the articles by Large and Pearson, which I have incorporated into the revised version of the paper. Also based on your recommendation, I have added a paragraph immediately after the introductory paragraph that defines the post-national imaginary in a theoretical framework.